# Enhancing Linear Bound Tightness in Neural Network Verification via Sampling-Based Underestimation

## Abstract

We present `PT-LiRPA` (Probabilistically Tightened LiRPA), a novel approach that enhances existing linear relaxation-based perturbation analysis (LiRPA) methods for neural network verification. `PT-LiRPA` combines LiRPA approaches with a sampling-based underestimation technique to compute probabilistically optimal intermediate bounds, resulting in tighter linear lower and upper bounds. Notably, we show that this approach preserves the soundness of verification results while significantly tightening the bounds for generic non-linear functions. Additionally, we introduce a new metric, $\Delta^*$, to quantify the tightness for LiRPA bounds and to bound the magnitude of the possible error in the sample-based overestimation, thus complementing the probabilistic bound of statistical results we use. Our empirical evaluation, conducted on several state-of-the-art benchmarks, including those from the International Verification of Neural Networks Competition, demonstrates that `PT-LiRPA` achieves higher or comparable verified accuracy with lower verification times. The significantly tighter bounds and better efficiency allow us to verify instances where state-of-the-art methods could not provide a specific answer.

## 1 Introduction

Deep neural networks (DNNs) and recently large language models (LLMs) have revolutionized various fields, from healthcare and finance to natural language processing, enabling unprecedented capabilities, for instance, in image recognition (O'Shea & Nash, 2015) and autonomous navigation (Tai et al., 2017). However, their opacity and vulnerability to the so-called "adversarial inputs" (Szegedy et al., 2013) raise significant concerns, particularly when they are deployed in safety-critical applications such as autonomous driving, medical diagnosis, or financial decision-making. Hence, developing methods to ensure that these models can be trusted, even in edge cases, is crucial. Provable safety guarantees involve formal verification (FV) techniques that mathematically ensure a system under a given amount of input perturbation will not produce harmful outcomes, offering a higher level of assurance than empirical testing alone.

Existing FV approaches tackle the problem in two main ways. The first solution consists of encoding the linear combinations and the non-linear activation functions of a DNN as a set of constraints for an optimization problem (Katz et al., 2017; Wu et al., 2024). The second method relies on *interval bound propagation* (IBP) (Lomuscio & Maganti, 2017; Gowal et al., 2018; Gehr et al., 2018) and consists of determining each neuron's reachable set, i.e., the lower and upper bound values until the output layer. However, due to the non-linear and non-convex nature of the DNN, computing the exact bounds of a neural network has been proven to be NP-hard (Katz et al., 2017). To address this challenging problem, a recent line of works called linear relaxation-based perturbation analysis (LiRPA) algorithms (Zhang et al., 2018; Xu et al., 2020b; Wang et al., 2021; Xu et al., 2020a) proposes a perturbation analysis based on a sound DNN linear relaxation. In detail, for a given DNN, the idea is to compute a linear relaxation of any non-linear activation function in the network. Thus for any possible input $x \in \mathcal{C}$ (with $\mathcal{C}$, for instance, an $\ell_\infty$ ball around the original input $x_0$), we can obtain two linear bounds for the output $f(x)$, an upper and lower bound, such that $\underline{f}(x) = \underline{a}^T x + \underline{c} \le f(x) \le \overline{f}(x) = \overline{a}^T x + \overline{c}$. Hence these approaches compute sound, over-

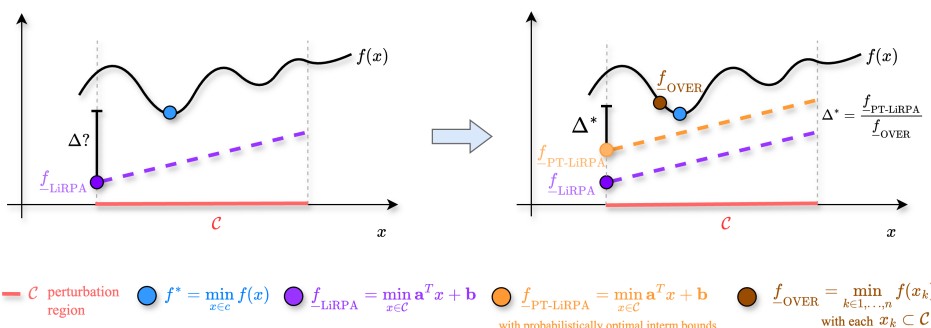

Figure 1: Overview of the proposed method in this work.

approximated linear bounds of the real minimum $f^* = \min\limits_{\boldsymbol{x} \in \mathcal{X}} f(\boldsymbol{x})$ (and maximum, respectively) that provide a conservative estimate of the system's behavior, thus covering all possible potential worst-case scenarios.[1] Even though this conservative approach enables to verify in a sound (and sometimes complete) way DNNs, it still presents two main limitations depicted in Figure 1 (left) that we aim to address in this paper. (i) The over-approximation approach $\underline{f}_{\text{LiRPA}}$ (in purple) could lead to loose bounds, thus preventing the possibility of answering the verification query. This problem is amplified in large networks, directly translating into scalability issues. (ii) To the best of our knowledge, no global optimal tightness guarantee ($\Delta$) of the computed bounds is provided for the approach of (Zhang et al., 2018; Xu et al., 2020b; Wang et al., 2021; Xu et al., 2020a). Recently, Biktairov & Deshmukh (2023) proposed an approach with an optimality criterion for the computation of tightness bound in terms of discrepancy volume between the lower (or upper) linear bound and the actual min (or max) value of the function. However, similar to other existing approaches (Liu et al., 2021), this procedure requires multiple invocations to a linear programming (LP) solver, which could result in prohibitive computational demand in the verification of large networks.

To tackle these challenging problems, in this work, we shift the focus from conventional over-approximation methods to examine the impact of provable probabilistic underestimation techniques. In detail, we propose `PT-LiRPA` (Probabilistically Tightened LiRPA), a novel approach for computing tighter linear lower and upper bounds by combining existing LiRPA methods with a sampling-based underestimation strategy. We first show that leveraging theoretical probabilistic guarantees (Wilks, 1942; Marzari et al., 2024), the overestimation of the actual minimum value of $f$—calculated using $n$ random input samples drawn uniformly from the perturbation region $\mathcal{C}$—could be incorrect for at most a countably small fraction of an indefinitely large additional sample set, with a predefined confidence level $\alpha$. Hence, we prove that by computing probabilistically optimal intermediate bounds in the DNN and combining them with any formal verification methods based on LiRPA, the soundness of the results is preserved for a specified confidence level (i.e., the method still yields valid overestimated lower and upper bounds with a confidence $\alpha$). Crucially, as also speculated by Xu et al. (2020b), having tighter intermediate reachable sets significantly tightens the final linear bounds, which directly translates into verification efficiency. The right side of Figure 1 depicts our idea: $\underline{f}_{\text{OVER}}$ the brown dot represents the overestimation of the min value of $f$, derived from a sampling-based approach within the perturbation region $\mathcal{C}$. By employing a similar procedure also to compute probabilistically optimal intermediate bounds (i.e., an under-estimation of the intermediate reachable sets) and by incorporating them in the linearization employed in the verification tools based on LiRPA, we can achieve significantly tighter linear lower bounds $\underline{f}_{\text{PT-LiRPA}}$ (in orange) which can lead to a more accurate verification result in less computational time. Crucially, we show that this approach allows us to provide precise answers even in instances where state-of-the-art approaches fail. To assess the improvement in the tightness of the new linear bounds, we provide a novel analytical formula $\Delta^*$ as a distance between $\underline{f}_{\text{OVER}}$ and $\underline{f}_{\text{PT-LiRPA}}$ to assess the global tightness of the bound relative to $f^*$. If $\Delta^* \to 0$, then both methods produce near-optimal bounds and provide a novel dual assurance: optimality guarantees for the LiRPA bounds and quali-

---

[1] For the sake of clarity and without loss of generality, we are only going to discuss the optimal lower bounding $\underline{f}(\boldsymbol{x})$. Similar considerations can also be applied when computing the upper bound $\overline{f}(\boldsymbol{x})$ with the necessary changes in computation.

tative insights on the magnitude of the possible error in the sample-based overestimation of the min of $f$, thus complementing the probabilistic bound of statistical results we employ. In detail, the result provided in Wilks (1942) quantitatively predicts how many new samples in a future indefinitely larger sample could be smaller than $\underline{f}_{\text{OVER}}$, calculated from the initial sample of size $n$. However, it does not specify how far apart these points could be from $\underline{f}_{\text{OVER}}$ (i.e., the distance). By employing $\Delta^*$, we can also provide, for the first time, a qualitative interpretation of this statistical outcome. This dual assurance represents a significant advancement in bounding non-linear functions, delivering more accurate results with reduced computational demand, thereby enhancing the reliability and scalability of safety verification for neural networks. In summary, the main contributions of the paper are:

- `PT-LiRPA`: a novel approach that combines existing over-approximation methods with sampling-based underestimation techniques with provable probabilistic guarantees to compute tighter linear bounds for deep neural networks (§3).

- A novel analytical formula, $\Delta^*$, to assess the global tightness of the computed bounds relative to the actual minimum function value, providing dual assurance of optimality for both over-approximated and probabilistically underestimated bounds (§3.2).

- A thorough empirical evaluation to assess the benefits of our approach. Crucially, our evaluation in different datasets and standard benchmarks of the international verification of neural networks competition (VNN-COMP) shows that `PT-LiRPA` obtains consistently similar or higher verified accuracy with respect to the original formal verification tools based on LiRPA counterpart (Zhang et al., 2018; Xu et al., 2020b; Wang et al., 2021; Xu et al., 2020a) while reducing verification time by several orders of magnitudes (§4). Crucially, we show that our method resolves instances that are considered "unknown" by previous state-of-the-art approaches.

## 2 PRELIMINARIES

For the sake of clarity in this section and all our work, we recall and simplify—when possible– main notation in related works on linear relaxation-based perturbation analysis (Xu et al., 2020b; Wang et al., 2021; Xu et al., 2020a).

### 2.1 NOTATION AND PROBLEM FORMULATION

Consider neural network classifier $f : \mathbb{R}^{d_0} \to \mathbb{R}$, with $d_0$ the input space dimension. We assume a model with $L$ layers ($L > 1$). In each layer we have weights $\boldsymbol{W}^{(i)} \in \mathbb{R}^{d_i \times d_{i-1}}$ and biases $\boldsymbol{b}^{(i)} \in \mathbb{R}^{d_i}$, for $i \in \{1, \ldots, L\}$. Given an input $\boldsymbol{x} \in \mathbb{R}^{d_0}$, we define the output of the neural network as the sequence of several linear and non-linear operations that produce: $f(\boldsymbol{x}) := \boldsymbol{z}^{(L)}(\boldsymbol{x})$ where $\boldsymbol{z}^{(i)}(\boldsymbol{x}) = \boldsymbol{W}^{(i)}\hat{\boldsymbol{z}}^{(i-1)}(\boldsymbol{x}) + \underline{\boldsymbol{b}}^{(i)}$ and $\hat{\boldsymbol{z}}^{(i)}(\boldsymbol{x}) = \sigma(\boldsymbol{z}^{(i)}(\boldsymbol{x}))$ is application of one arbitrary (non-)linear activation function with $\hat{\boldsymbol{z}}^{(0)}(\boldsymbol{x}) = \boldsymbol{z}^{(0)}(\boldsymbol{x}) = \boldsymbol{x}$. We define we the symbol $z_j^{(i)}(\boldsymbol{x})$ and $\hat{z}_j^{(i)}(\boldsymbol{x})$ the pre and post-activation values of the $j$-th neuron in the $i$-th layer, respectively (see Figure 3 in Appendix B). In this work, we consider Rectified Linear Unit (ReLU) as an activation function which is the most employed in the literature verification works (Xu et al., 2020b; Wang et al., 2021), but the soundness of the proposed approach still holds with different non-linear scalar functions studied in literature such as Tanh, Sigmoid, GeLU, etc. For practical purposes and without loss of generality, we observe that it is possible to assume that the network has a single output node on whose we can verify the desired safety/robustness property. We can enforce this condition for networks that do not satisfy this assumption by adding one layer and encoding, for instance, the robustness property we aim to verify in a single output node as a margin between logits, which produces a positive output only if the correct label is predicted (Liu et al., 2021; Wang et al., 2021). Hence, we can define the robustness verification problem of deep neural networks as follows.

Given an input perturbation set $\mathcal{C} = \{\boldsymbol{x}| \ ||\boldsymbol{x} - \boldsymbol{x}_0||_\infty \leq \epsilon\}$, i.e., with $\mathcal{C}$ as an $\ell_\infty$ ball around an original input $\boldsymbol{x}_0$, we aim to find, if exists, an input $\boldsymbol{x} \in \mathcal{C}$ such that $f(\boldsymbol{x}) < 0$, thus resulting in a violation of the property. If $f(\boldsymbol{x}) \geq 0 \ \forall \boldsymbol{x} \in \mathcal{C}$, we say $f(\boldsymbol{x})$ is robust (or verified) to all the possible

input perturbations in $\mathcal{C}$. A possible way to prove the property is to solve the optimization problem in terms of $\min_{\boldsymbol{x} \in \mathcal{C}} f(\boldsymbol{x})$ and by checking if the result is positive. Formally:

**Definition 1** (*Robustness verification problem*).

**Input**: *A tuple* $\mathcal{T} = \langle f, \mathcal{C} \rangle$.

**Output**: Robust $\iff \min_{\boldsymbol{x} \in \mathcal{C}} f(\boldsymbol{x}) := z^{(L)}(\boldsymbol{x}) \geq 0$.

However, given the non-convex transformation imposed by $\hat{z}^{(i)}$, i.e., by the non-linear activation functions, Def. 1 presents a non-convex NP-hard optimization problem to solve (Katz et al., 2017). To address this problem, (in)complete verifiers usually relax the DNNs' non-convexity to obtain over-approximate sound lower $\underline{f}$ and upper $\overline{f}$ bounds of $f$. If $\underline{f} \geq 0$, then also $\underline{f}^*$, i.e., the real minimum value of $f$ will be positive, and similarly if $\overline{f} < 0$ than also $\overline{f}^* < 0$. In both these situations, we can return a provable result. In the last situation, namely if $\underline{f} < 0 < \overline{f}$, we cannot provide an answer, and we typically have to proceed with a branch and bound (BaB) (Bunel et al., 2018). More specifically, many FV tools firstly recursively divide the original verification problem into smaller subdomains either, for instance, dividing the perturbation region (Wang et al., 2018) or splitting ReLU neurons into positive/negative linear domains (Bunel et al., 2020). Secondly, they bound each subdomain with specialized (incomplete) verifiers, typically linear programming (LP) solvers (Ehlers, 2017), which can fully encode neuron split constraints. The verification process ends once either we verify all the subdomains of this searching tree or we find a single counterexample where $\overline{f} < 0$. Even though LP-verifiers are mainly used in complete FV tools, recent LiRPA-based approaches Xu et al. (2020b); Wang et al. (2021) show how to solve an optimization problem that is equivalent to the costly LP-based methods with neuron split constraints while maintaining the efficiency of bound propagation techniques significantly outperforming LP-verification time thanks to GPU's acceleration.

## 2.2 Linear Relaxation-based Perturbation Analysis (LiRPA) approaches

To produce linear bounds of a DNN, LiRPA approaches (Zhang et al., 2018; Singh et al., 2019; Xu et al., 2020a;b; Wang et al., 2021) propose to resolve non-linearity in the neural network by computing linear relaxation of each non-linear unit. The high-level idea is to compute the linear bounds of each neuron in the DNN, for instance, all the ReLU nodes, to express a linear relation between layers. In detail, using bound propagation, we first compute a lower and upper bound for each neuron $l_j^{(i)} \leq z_j^{(i)} \leq u_j^{(i)}$. A ReLU

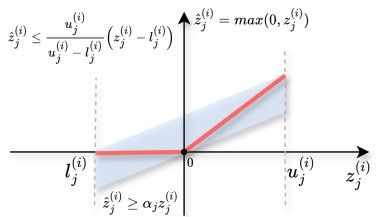

Figure 2: Linear relaxation for ReLU($z_j^{(i)}$)

node, $\hat{z}_j^{(i)} = \max(0, z_j^{(i)})$, is considered *"unstable"* if its pre-activated bounds are $u_j^{(i)} > 0 > l_j^{(i)}$ and can be linearized as depicted in Figure 2. In the other cases, is either considered *"active"* if $l_j^{(i)} \geq 0$ or *"inactive"* if $u_j^{(i)} \leq 0$. Once linear bounds are established across all neurons, two propagation methods are typically employed: forward and backward. In forward propagation, the linear bounds for each neuron are expressed in terms of the input and propagated layer by layer until the output is reached. In backward propagation, we start from the output and propagate the bounds backward to earlier layers until we can express a linear relation between input and output.

To improve the tightness of the bounds, Xu et al. (2020a) suggest using a refined backward propagation based on the results of a preceding forward pass. In detail, once we have the intermediate reachable sets, we start by defining the new linear dependency of each layer $i$ with respect to the previous one $i - 1$ using a vector $\boldsymbol{A}^{(i)} = \boldsymbol{A}^{(i-1)} \underline{\boldsymbol{D}}^{(i)} \boldsymbol{W}^{(i)}(\boldsymbol{x})$, with $i \in \{1, \dots, L\}$ and $\boldsymbol{A}^{(L)} = I$, $\boldsymbol{A}^{(L-1)} = \boldsymbol{w}^T$, assuming a single output node. In detail, $\underline{\boldsymbol{D}}^{(i)}$ is a diagonal matrix that expresses the linear relaxation of the $i$-th non-linear layer $\hat{\boldsymbol{z}}^{(i)}$. Each diagonal coefficient $\underline{\boldsymbol{D}}_{j,j}$ of the matrix is based on the preactivated reachable set of the node $z_j^{(i)}$, computed in the forward propagation and on

the sign of $\boldsymbol{A}_j^{(i-1)}$, i.e., the $j$-th element in the vector that represents the linearization of the previous layer.

Zhang et al. (2018) show the given two vectors $\boldsymbol{A}, \boldsymbol{v} \in \mathbb{R}^d$, where $\boldsymbol{v}$ is the pre-activated ReLU vector, with $\boldsymbol{l} \leq \boldsymbol{v} \leq \boldsymbol{u}$ (element-wise) and $\boldsymbol{A}$ is the vector of the linear bounds coefficients entering in the ReLU layer, we have:

$$\boldsymbol{A}^T ReLU(\boldsymbol{v}) \geq \boldsymbol{A}^T(\underline{\boldsymbol{D}}\boldsymbol{v} + \underline{\boldsymbol{b}}) \tag{1}$$

where $\underline{\boldsymbol{D}}$ is a diagonal matrix (thus, we omit the subscripts $(j,j)$ to denote the elements of the diagonal in the notation) and $\underline{\boldsymbol{b}}$ the biases that linearize each specific ReLU node[2] defined as:

$$\underline{\boldsymbol{D}} = \begin{cases} 1 & \boldsymbol{l}_j \geq 0, \\ 0 & \boldsymbol{u}_j \leq 0, \\ \alpha_j & \boldsymbol{u}_j > 0 > \boldsymbol{l}_j \text{ and } \boldsymbol{A}_j \geq 0, \\ \frac{\boldsymbol{u}_j}{\boldsymbol{u}_j - \boldsymbol{l}_j} & \boldsymbol{u}_j > 0 > \boldsymbol{l}_j \text{ and } \boldsymbol{A}_j < 0 \end{cases} \qquad \underline{\boldsymbol{b}} = \begin{cases} 0 & \boldsymbol{l}_j > 0 \text{ or } \boldsymbol{u}_j \leq 0, \\ 0 & \boldsymbol{u}_j > 0 > \boldsymbol{l}_j \text{ and } \boldsymbol{A}_j \geq 0, \\ -\frac{\boldsymbol{u}_j \boldsymbol{l}_j}{\boldsymbol{u}_j - \boldsymbol{l}_j} & \boldsymbol{u}_j > 0 > \boldsymbol{l}_j \text{ and } \boldsymbol{A}_j < 0. \end{cases}$$

Thus they prove that given an $L$-layer ReLU DNN $f(x) : \mathbb{R}^{d_0} \to \mathbb{R}$ with weights $\mathbf{W}^{(i)}$, biases $\mathbf{b}^{(i)}$, pre-ReLU bounds $\mathbf{l}^{(i)} \leq \boldsymbol{z}^{(i)} \leq \mathbf{u}^{(i)}$ and an input constraint $\boldsymbol{x} \in \mathcal{C}$, it holds

$$\min_{\boldsymbol{x} \in \mathcal{C}} f(\boldsymbol{x}) \geq \min_{\boldsymbol{x} \in \mathcal{C}} \boldsymbol{a}_{\text{LiRPA}}^T(\boldsymbol{x}) + c_{\text{LiRPA}}. \tag{2}$$

Where $\boldsymbol{a}_{\text{LiRPA}}^T$ and $c_{\text{LiRPA}}$ are the coefficients of the linear equation for the lower bound of $f(x)$. We provide further details and an explanatory example of linear bounds computation for a toy DNN in Appendix B.

## 2.3 RELATED WORK

In recent years, significant research has been dedicated to increasing the quality of linear bounds of the most popular activation functions, such as ReLU, and more general activation functions. More specifically, (Xu et al., 2020a) proposes a framework for deriving and computing near-optimal sound bounds with linear relaxation-based perturbation analysis for neural networks. This framework is the base of all the most famous state-of-the-art formal verification tools such as CROWN (Zhang et al., 2018), $\alpha$-CROWN (Xu et al., 2020b), $\beta$-CROWN (Wang et al., 2021), the top performer on last years VNN-COMP (Müller et al., 2022; Brix et al., 2023). Recently, different approaches have tried to incorporate a sampling-based approach to enhance either the linear relaxation of arbitrary non-linear functions (Paulsen & Wang, 2022; Biktairov & Deshmukh, 2023) or the verification process (Balunovic et al., 2019). For instance, (Paulsen & Wang, 2022) proposed a method synthesizing linear bounds for arbitrary complex activation functions, such as GeLU (Hendrycks & Gimpel, 2016) and Swish (Ramachandran et al., 2017), by combining a sampling technique with an LP solver to synthesize candidate lower and upper bound coefficients and then certified the final result via SMT solvers (Gao et al., 2013). However, no tightness optimality guarantees are returned for the bounds computed. To address such an issue, (Biktairov & Deshmukh, 2023) presented a combination of an efficient sampling-based approach and linear programming solvers for finding linear bounds arbitrarily close to optimum in terms of tightness for Lipschitz-continuous functions. Unlike LinSyn, their approach provides optimality guarantees based on LP verification for the generated bounds and does not heavily rely on using the SMT solver, resulting in superior running time performance. However, they still require an LP solver to provide optimality guarantees.

In contrast, the scope of this paper is to provide a method to probabilistically enhance the tightness of existing LiRPA linear bounds without relying on any LP or SMT solvers. Hence, our methodology and comparative analysis will be predominantly based on the available auto_LiRPA (Xu et al., 2020a), which recently has also incorporated the improvements of (Paulsen & Wang, 2022; Biktairov & Deshmukh, 2023), and $\alpha, \beta-$CROWN (Zhang et al., 2018; Xu et al., 2020b; Wang et al., 2021) frameworks. This allows us to effectively derive and assess the tightness of our linear bounds approach with respect to recent state-of-the-art approaches also employed in the VNN-COMP.

---

[2]We do not report for clarity of reading the superscript $(i)$ on $\boldsymbol{D}$ and $(i + 1)$ on $\boldsymbol{A}_j$. We use the complete notation in Appendix B.

## 3 PROBABILISTICALLY TIGHTENED LiRPA VIA UNDERESTIMATION

In this section, we present all the theoretical and practical components of our `PT-LiRPA` approach to computing tightened bounds with probabilistic guarantees on the optimality of the result returned. As previously introduced, our approach is based on two main components: probabilistically optimal intermediate bounds and a tight overestimation of $f^*$.

We start by exploiting statistical results known as *Statistical Prediction of Tolerance Limits* (Wilks, 1942) to derive provable probabilistic guarantees on the underestimated reachable sets computed from a sampling-based approach. Let $z_j^{(i)}$ be a node in a neural network with pre-activation values computed from a uniform sample of $n$ points drawn from a continuous perturbation set on interest $\mathcal{C}$. We compute its pre-activated bounds as:

$$\bar{l}_j^{(i)} = \min_{k=1,\dots,n} z_j^{(i)}(x_k); \qquad \underline{u}_j^{(i)} = \max_{k=1,\dots,n} z_j^{(i)}(x_k),$$

i.e., the minimum and maximum value obtained from of the propagation of $n$ random points in that specific neuron $z_j^{(i)}$. Notably, since we perform one single propagation on $n$ random inputs, $\bar{l}_j^{(i)}$ and $\underline{u}_j^{(i)}$ preserve the soundness and interdependence between the layers. Nonetheless, since we are using a sample-based approach with high probability, we are underestimating $[l^{*(i)}_j, u^{*(i)}_j]$, i.e., the real lower and upper bound for that node. In fact, we have the following proposition.

**Proposition 1** (Underestimation of sampling-based approaches). *Let $z_j^{(i)}$ the $j$-th neuron in the $i$-th layer and $l^{*(i)}_j$, $u^{*(i)}_j$ the true lower and upper bounds of $z_j^{(i)}$, respectively. Then for $\bar{l}_j^{(i)} = \min_{k=1,\dots,n} z_j^{(i)}(x_k)$ and $\underline{u}_j^{(i)} = \max_{k=1,\dots,n} z_j^{(i)}(x_k)$ the lower and upper bounds of $z_j^{(i)}$ computed using a sampling-based approach, it necessary holds: $\bar{l}_j^{(i)} \geq l^{*(i)}_j; \quad \underline{u}_j^{(i)} \leq u^{*(i)}_j$.*

By choosing a sample size based on the results of Wilks (1942), we can achieve a quantitative correctness result in terms of probability $\alpha$ that our estimate of the intermediate reachable set holds for at least a fixed (chosen) fraction $R$ of a further possibly infinitely large sample of inputs from the same perturbation set $\mathcal{C}$. Crucially, this statistical result does not require any knowledge of the probability distribution governing our function of interest and thus also applies to our setting.

**Lemma 1** (Probabilistically optimal pre-activated intermediate bounds). *Let $n$ the number of samples employed in the computation and the interval $[\bar{l}_j^{(i)}, \underline{u}_j^{(i)}]$, where $\bar{l}_j^{(i)}$ and $\underline{u}_j^{(i)}$ are the minimum and maximum pre-activation values observed in the sample, respectively. Fix $R \in (0, 1)$, then for any further possibly infinite sequence of samples from $\mathcal{C}$, the probability that $[\bar{l}_j^{(i)}, \underline{u}_j^{(i)}]$ is incorrect for more than $1 - R$ of points is at most $1 - \alpha$, with $\alpha = n \cdot \int_R^1 x^{n-1} \, dx = (1 - R^n)$.*

Hence, following lemma 1, by selecting a desired confidence level $\alpha$, and a fraction $R$, we can derive the number of samples necessary to obtain the provable probabilistic guarantees on the intermediate bounds computed. Notably, we have that for $n \geq \frac{\ln(1-\alpha)}{\ln(R)}$ samples used to compute $[\bar{l}_j^{(i)}, \underline{u}_j^{(i)}]$, with probability $\alpha$ at most a fraction $(1 - R)$ of points in an indefinitely larger future sample could fall outside that reachable set.

We now prove that, by utilizing these probabilistically optimal underestimation techniques to compute intermediate bounds in the DNN and combining them with any LiRPA formal verification methods, the soundness of the results in terms of lower (and upper, respectively) bound of $f$ returned is probabilistically preserved, with a predefined confidence level $\alpha$. We start by showing that the soundness of the relaxation of the ReLU layers using any LiRPA approaches is still probabilistically preserved using `PT-LiRPA`.

**Lemma 2** (ReLU Layer Relaxation using `PT-LiRPA`). *Fix $\alpha, R \in (0, 1)$. Given two vectors $\boldsymbol{A}^*, \boldsymbol{v} \in \mathbb{R}^d$, where $\boldsymbol{v}$ is the pre-activated ReLU vector, with $\bar{\boldsymbol{l}} \leq \boldsymbol{v} \leq \underline{\boldsymbol{u}}$ (element-wise) obtained from a sampled of $n \geq \frac{\ln(1-\alpha)}{\ln(R)}$ samples and $\boldsymbol{A}^*$ is the vector of the linear bounds coefficients of the previous ReLU layer (computed with the probabilistically optimal intermediate bounds), with a confidence $\geq \alpha$ it holds:*

$$\boldsymbol{A}^{*T} ReLU(\boldsymbol{v}) \geq \boldsymbol{A}^{*T}(\underline{\boldsymbol{D}}^* \boldsymbol{v} + \underline{\boldsymbol{b}}^*) \tag{3}$$

*where*

$$\underline{\boldsymbol{D}}^* = \begin{cases} 1 & \bar{l}_j \geq 0, \\ 0 & \underline{\boldsymbol{u}}_j \leq 0, \\ \alpha_j & \underline{\boldsymbol{u}}_j > 0 > \bar{l}_j \text{ and } \boldsymbol{A}_j^* \geq 0, \\ \frac{\underline{\boldsymbol{u}}_j}{\underline{\boldsymbol{u}}_j - \bar{l}_j} & \underline{\boldsymbol{u}}_j > 0 > \bar{l}_j \text{ and } \boldsymbol{A}_j^* < 0 \end{cases} \qquad \underline{\boldsymbol{b}}^* = \begin{cases} 0 & \bar{l}_j > 0 \text{ or } \underline{\boldsymbol{u}}_j \leq 0, \\ 0 & \underline{\boldsymbol{u}}_j > 0 > \bar{l}_j \text{ and } \boldsymbol{A}_j^* \geq 0, \\ -\frac{\underline{\boldsymbol{u}}_j \bar{l}_j}{\underline{\boldsymbol{u}}_j - \bar{l}_j} & \underline{\boldsymbol{u}}_j > 0 > \bar{l}_j \text{ and } \boldsymbol{A}_j^* < 0. \end{cases}$$

The proof is reported in Appendix A. As a direct implication of this result, we can show that the linear lower and upper bounds computed using `PT-LiRPA` still remain probabilistically valid.

**Lemma 3** (`PT-LiRPA` lower bound). *Given an L-layer ReLU DNN $f(x) : \mathbb{R}^{d_0} \rightarrow \mathbb{R}$ with weights $\boldsymbol{W}^{(i)}$, biases $\boldsymbol{b}^{(i)}$, pre-ReLU bounds $\bar{\boldsymbol{l}}^{(i)} \leq \boldsymbol{z}^{(i)} \leq \underline{\boldsymbol{u}}^{(i)}$ and an input constraint $x \in \mathcal{C}$, it holds with probability $\geq \alpha$*

$$\min_{\boldsymbol{x} \in \mathcal{C}} f(\boldsymbol{x}) \geq \min_{\boldsymbol{x} \in \mathcal{C}} \boldsymbol{a}_{PT\text{-}LiRPA}^T(\boldsymbol{x}) + \boldsymbol{c}_{PT\text{-}LiRPA}$$

*Proof.* The proof directly follows from our Lemma 2 and the derivations of Zhang et al. (2018). ☐

### 3.1 `PT-LiRPA` FRAMEWORK

Based on the theoretical results of our approach, we now present in Algorithm 1 the `PT-LiRPA` approach for the verification process. For the sake of clarity and without loss of generality, we present the procedure applied to the parallel BaB as shown for the optimized LiRPA approach proposed in (Xu et al., 2020b).

---

**Algorithm 1:** `PT-LiRPA` on parallel BaB

**Input** : a DNN $f$, a region $\mathcal{C}$, sample size $n$, confidence parameter $R$, batch size $m$.
**Output** : *robust/not-robust* with a confidence $\geq \alpha = 1 - R^n$

1   *interm_bounds* $\leftarrow$ `get_interm_bounds`$(f, \mathcal{C}, n)$
2   $\underline{f}_\mathcal{C}, \overline{f}_\mathcal{C} \leftarrow$ `LiRPA`$(f, \mathcal{C}, interm\_bounds)$
3   $\mathcal{B} \leftarrow \{(\underline{f}_\mathcal{C}, \overline{f}_\mathcal{C})\}$
4   **while** $\mathcal{B} \neq \emptyset$ **do**
5     $\mathcal{C}_1, \ldots, \mathcal{C}_m \leftarrow$ `split`$(\mathcal{B}, m)$
6     *interm_bounds*$_{\mathcal{C}_1, \ldots, \mathcal{C}_m} \leftarrow$ `get_interm_bounds`$(f, [\mathcal{C}_1, \ldots, \mathcal{C}_m], n)$
7     $(\underline{f}_{\mathcal{C}_1}, \overline{f}_{\mathcal{C}_1}), \ldots, (\underline{f}_{\mathcal{C}_m}, \overline{f}_{\mathcal{C}_m}) \leftarrow$ `LiRPA`$(f, (\mathcal{C}_1, \ldots, \mathcal{C}_m), interm\_bounds_{\mathcal{C}_1, \ldots, \mathcal{C}_m})$
8     $\mathcal{B} \leftarrow \mathcal{B} \cup \mathcal{B} \setminus$ `get_robust_domain`$((\underline{f}_{\mathcal{C}_1}, \overline{f}_{\mathcal{C}_1}), \ldots, (\underline{f}_{\mathcal{C}_m}, \overline{f}_{\mathcal{C}_m}))$
9     **if** $\exists \overline{f}_{\mathcal{C}_i} < 0$ *in* $\mathcal{B}$ **then**
10       **return** *not robust*
11     **end**
12   **end**
13   **return** *robust*

---

Given a DNN $f$ and a region of interest $\mathcal{C}$ the verification process of state-of-the-art verification tools typically involves a projected gradient descent (PGD) attack (Madry et al., 2018), which is employed either before starting or during the BaB process to search for potential adversarial input in the region under consideration. If no adversarial is found, the BaB process starts. For a given sample size $n$ and confidence parameter $R$, we first compute probabilistically optimal intermediate bounds using `get_interm_bounds` method, and then use these bounds in the linear bounds computation on any existing `LiRPA` approach. We store the resulting bounds $\underline{f}$ and $\overline{f}$ for the region $\mathcal{C}$, namely $\underline{f}_\mathcal{C}$ and $\overline{f}_\mathcal{C}$ in a set $\mathcal{B}$ of unverified regions (lines 1-3). We then start the BaB process by splitting using the `split` method the original region from $\mathcal{B}$ into $m$ sub-regions (line 5). Notably, we can perform the parallel selection and splitting into sub-domains using information on unstable ReLU nodes, as shown in (Bunel et al., 2020; Wang et al., 2021), or just on the perturbation region $\mathcal{C}_i$ (Wang et al., 2018). Once we have the new sub-domains, we recompute the intermediate reachable sets in parallel and use these bounds for the new computation of the linear lower and upper bounds for each sub-region (lines 6-7). Finally, we update $\mathcal{B}$ with the resulting unverified sub-domains from the procedure `get_robust_domain` (line 8). The verification process continues until either $\mathcal{B}$ is empty, returning a *robust* answer, or we find an adversarial configuration, i.e., there is at least a single sub-domain $\mathcal{C}_i$ that presents $\overline{f} < 0$, thus returning *not robust* as the answer (lines 9-13).

Crucially since `PT-LiRPA` could overestimate the lower bound (and respectively underestimate the upper bound of a region $C_i$), we can perform either a sample-based or a PGD attack in the `get_robust_domain` procedure, to empirically asses whether we wrongly deemed a region as robust. Following Lemma 3, if no adversarial is found with this further check, we can state that with a confidence $\geq 1 - R^n$ in that region $C_i$, the DNN is robust.

### 3.2 A NOVEL ANALYTIC FORMULA TO ASSESS THE TIGHTNESS OF LINEAR BOUNDS

In this section, we derive a novel analytical formula to estimate the tightness of the bounds computed using the `PT-LiRPA` approach. This formula provides a means to evaluate how closely the overapproximated or underestimated bounds align with the actual minimum values of the neural network function $f$.

Given a perturbation region $C$ of interest and the minimum value of the neural network function $f^* = \min_{\boldsymbol{x} \in C} f(\boldsymbol{x})$, we define $\underline{f}_{\text{OVER}} = \min_{k=\{1,\ldots,n\}} f(x_k)$, where each $x_k$ sampled is in $C$, the overapproximated estimate value of $f^*$ computed using a sampling-based method. By knowing the value of $f^*$ we can estimate the exact tightness of $\underline{f}_{\text{OVER}}$, by measuring the relative difference between the overapproximated minimum and the true minimum, normalized by the magnitude of $f^*$. Specifically, we can write this formula:

$$\Delta = \frac{\underline{f}_{\text{OVER}} - f^*}{|f^*| + \varepsilon} \tag{4}$$

We note that if $f^*$ tends towards 0, $\Delta$ will wrongly return an indefinitely larger result with respect to the actual ratio. To address such an issue, we sum a small quantity $\varepsilon > 0$ to avoid division for minimal values and still preserve the correctness of the formula. Nonetheless, in many practical scenarios, computing $f^*$ of the neural network function is hard or even potentially unfeasible. To address this, we can derive an upper bound on the tightness estimate $\Delta$ by replacing the true minimum $f^*$ with our probabilistically optimal lower bound of $f$.

**Corollary 1** (Analytic formula for probabilistic global tightness). *Let $C$ be a perturbation region of interest, and $f$ be a neural network. Given $\underline{f}_{\text{OVER}} = \min_{k=\{1,\ldots,n\}} f(x_k)$ the overestimation of minimum value of $f$ and $\underline{f}_{\text{PT-LiRPA}} = \min_{\boldsymbol{x} \in C} \boldsymbol{a}_{\text{PT-LiRPA}}^T(\boldsymbol{x}) + \boldsymbol{c}_{\text{PT-LiRPA}}$ the probabilistically optimal lower bound of $f$, a valid upper bound of Equation 4 is given by:*

$$\Delta^* = \frac{\underline{f}_{\text{OVER}} - \underline{f}_{\text{PT-LiRPA}}}{\min\left(|\underline{f}_{\text{OVER}}|, |\underline{f}_{\text{PT-LiRPA}}|\right) + \varepsilon} \tag{5}$$

Where $\underline{f}_{\text{PT-LiRPA}}$ is the probabilistic optimal lower bound of the neural network function, which is obtained by integrating the probabilistic optimal intermediate bounds into any existing LiRPA method. The upper bound formula $\Delta^*$ derivation follows from the principles of linear relaxation and probabilistic underestimation outlined in §3. In detail, by considering the probabilistic nature of the lower bounds $\underline{f}_{\text{PT-LiRPA}}$ derived through sampling-based approaches, we know that $\underline{f}_{\text{PT-LiRPA}} \leq f^*$ with a confidence $\geq \alpha$ (as guaranteed by Lemma 3). Hence, the difference $\underline{f}_{\text{OVER}} - \underline{f}_{\text{PT-LiRPA}}$ serves as an overestimation of the difference $\underline{f}_{\text{OVER}} - f^*$. To normalize this difference, we divide by the minimum magnitude of $\underline{f}_{\text{OVER}}$ or $\underline{f}_{\text{PT-LiRPA}}$, ensuring that the upper bound $\Delta^*$ remains a meaningful estimate even when the exact value of $f^*$ is unknown. Once again, we add a small quantity $\varepsilon > 0$ to avoid division for too small values. This upper bound provides a practical and theoretically grounded method to assess the tightness of the bounds obtained through the `PT-LiRPA` or any LiRPA approaches.

## 4 EXPERIMENTAL EVALUATION

In this section, we empirically validate the effectiveness and the correctness of theoretical results of the `PT-LiRPA`. In detail, we present two sets of experiments to answer the following questions: (i) What is the impact of probabilistically optimal intermediate bounds on the computation of the

| Results w/ MIP (Tjeng et al., 2017) as ground truth | | | |
|---|---|---|---|
| Method | Mean $\ell_2$ norm | % tighter | Mean $\Delta^*$ |
| CROWN | 0.84 | - | 0.23 |
| **CROWN w/ PT-LiRPA** | **0.49** | **42%** | **0.12** |
| $\alpha$-CROWN | 0.42 | - | 0.1 |
| $\alpha$-**CROWN w/ PT-LiRPA** | **0.28** | **31.4%** | **0.07** |
| PT-OVER | 0.002 | - | - |

| Results w/ Powell (Powell, 1989) as ground truth | | | |
|---|---|---|---|
| Method | Mean $\ell_2$ norm | % tighter | Mean $\Delta^*$ |
| CROWN | 4.14 | - | 2.56 |
| **CROWN w/ PT-LiRPA** | **0.20** | **95%** | **0.04** |
| $\alpha$-CROWN | 0.82 | - | 0.69 |
| $\alpha$-**CROWN w/ PT-LiRPA** | **0.13** | **84.3%** | **0.03** |
| PT-OVER | 0.004 | - | - |

Table 1: Results with MIP as ground truth    Table 2: Results without MIP as ground truth

lower bound of an arbitrary function $f$? (ii) How accurate is the analytic formula for estimating the tightness of the bound if the true minimum of the function is unknown? (iii) How much does the use of `PR-LiRPA` impact the lower bounds and verification process in realistic models/benchmarks? We provide the code, trained models, and comprehensive instructions for reproducing our results in the supplementary material.

**Comparison of `auto_LiRPA` and `PT-LiRPA` linear bounds.** To answer the first two questions, we compare the quality of the linear lower bounds of `PT-LiRPA` and `auto_LiRPA` on a synthetic dataset of 2000 models. In the first test, we consider 500 randomly generated models from 5 random seeds, such that a computation of the real min of the neural network is achievable by employing Mixed Integer Programming (MIP) (Tjeng et al., 2017), implemented in `auto_LiRPA`. To test the scalability and effectiveness of our approach, in this second experiment, we consider other 1500 random models with different non-linear activation functions, such as Tanh and Sigmoid, beyond ReLU and larger models where MIP cannot be employed. In both the experiments for `PT-LiRPA` approaches, we set $\alpha = 0.9999$ (i.e., the answer is correct with a confidence $\geq 99.99\%$) and the fraction of tolerance error $1 - R = 0.001$. Following Lemma 1, this setting required a sample size of $n \geq 9205$. Hence, we use a sample size of $10k$ random input from the input region of interest in these first two experiments to compute the probabilistically optimal intermediate bounds.

For each experiment, we compute the lower bound of the models using four different bound propagations strategies, namely CROWN (Zhang et al., 2018), $\alpha$-CROWN (Xu et al., 2020b), and their corresponding enhanced implementation in our `PT-LiRPA` with probabilistically optimal intermediate bounds. We then compare the final linear bounds with the MIP result in terms of mean $\ell_2$ norm distance, if available, or with the Powell (Powell, 1989) algorithm–to have still an intuition of the tightness of the computed bounds in terms of $\ell_2$ norm. Additionally, we compute the mean over all the models tested of our novel analytic formula $\Delta^*$, without relying on the MIP result, to show the relation between exact $\ell_2$ norm distance (when available) and the ratio between linear bounds computed with different approaches. Notably, our empirical results on 2000 models show that `PT-LiRPA` always produced a valid lower bound of $f(x)$, comparing the minimum discovered by our approach and the one returned by MIP and Powell. In Appendix C, we report all the details regarding the model tested and the hyperparameters used.

Results in Table 1 and 2 show that, in general, `PT-LiRPA` can improve the tightness of the linear bounds by at least 30% on smaller models, reaching up to more than 80% on larger models. We do not provide the computational times for the two approaches, as they are comparable. In fact, compared to any LiRPA method, `PT-LiRPA` only adds the requirement of a single forward pass of $n$ random inputs, storing all intermediate results—an operation that can be efficiently performed in batches using GPU acceleration as highlighted in our ablation study in Appendix D.

**Impact of `PT-LiRPA` in the formal verification process.** To answer the last question, we integrate our `PT-LiRPA` in $\alpha, \beta$-CROWN (Xu et al., 2020b; Wang et al., 2021) and perform a final experiment on different benchmarks of the VNN-COMP 2022 and 2023 (Müller et al., 2022; Brix et al., 2023). This set of experiments aims to confirm our hypothesis regarding the effectiveness of having tighter intermediate bounds for verification purposes. In detail, our intuition is that with tighter intermediate bounds, we can achieve more precise final reachable sets, which reduces the cases where the verification approach can not make a decision and must resort to a split in the BaB process. Hence, by reducing these situations, we can achieve faster verification results.

Table 3 reports our results, where we consider an increased difficulty for the verification process.[3] We start with the simpler benchmark *ACASxu* (Julian et al., 2016; Katz et al., 2017), and we test

---

[3]We refer the interested readers to further detail in the benchmarks used to Appendix C and to the final report of the VNN-COMP available here (Müller et al., 2022; Brix et al., 2023)

| Results on VNN-COMP 2022-2023 benchmarks | | | | | | |
|---|---|---|---|---|---|---|
| Benchmark | Method | Verified accuracy | #safe/unsat | #unsafe/sat | #unkwown | Tot verification time |
| *ACASxu prop. 3* | $\alpha,\beta$-CROWN | 93.33% | 42 | 3 | 0 | 26s |
| | $\alpha,\beta$-CROWN w/ `PT-LiRPA` | 93.33% | 42 | 3 | 0 | **16.37s** |
| *tllVerifyBench* | $\alpha,\beta$-CROWN | 46.875% | 15 | 17 | 0 | **90.2s** |
| | $\alpha,\beta$-CROWN w/ `PT-LiRPA` | 46.875% | 15 | 17 | 0 | 92s |
| *CIFAR_biasfield* | $\alpha,\beta$-CROWN | 95.83% | 69 | 1 | 2 | 1553.5s |
| | $\alpha,\beta$-CROWN w/ `PT-LiRPA` | **98.61%** | **71** | 1 | **0** | **408.7s** |
| *CIFAR_tinyimagenet* | $\alpha,\beta$-CROWN | 62.5% | 15 | 3 | 6 | 1429.6s |
| | $\alpha,\beta$-CROWN w/ `PT-LiRPA` | **87.5%** | **21** | 3 | **0** | **425.6s** |

Table 3: Results on VNN-COMP 2022-2023 benchmarks. Results in **green bold** report the best-resulted method in terms of verified accuracy (% sat instances/all instances) and total verification time for the specific benchmark tested.

property 3. This property is particularly interesting as it holds for 42 of the 45 models tested, thus allowing us to verify the improvement in terms of time and verification accuracy. In the first row of Table 3, we can notice that $\alpha, \beta$-CROWN enhanced with `PT-LiRPA` achieves the same verified accuracy in less verification time, thus confirming our intuition. Interestingly, we observe that tighter bounds are not always beneficial in general. Specifically, in cases where a PGD attack succeeds despite loose bounds, using tighter bounds does not lead to further improvements. Additionally, in some scenarios, less accurate bounds from vanilla LiRPA methods could be quickly refined by BaB, still resulting in efficient verification time. This is exemplified by the *tllVerifyBench* experiments, where `PT-LiRPA` produced tighter intermediate bounds but achieved the same verified accuracy with a minor overhead in bounds computation.

Finally, we test our `PT-LiRPA` approach in more challenging verification benchmarks such as *CIFAR_biasfield* and *CIFAR_tinyimagenet*. Both these benchmarks are image-based verification tasks and thus allow us to show the scalability of the proposed approach. Before initiating the verification process with `PT-LiRPA`, we conduct a preliminary ablation study to evaluate the impact of different sample sizes on intermediate bounds computation. Specifically, in *CIFAR_biasfield* benchmark, our results in Appendix D indicate that stable intermediate bounds—defined as maintaining a consistently small pre-defined distance from the reference bounds computed with a confidence level of $\alpha \geq 99.999\%$ and $R = 0.00001$—can be achieved using between $250k$ and $350k$ samples. Hence we use $350k$ to compute intermediate bounds in the verification approach. Crucially, in these two last benchmarks, we obtain huge improvements in verification results with respect to $\alpha, \beta$-CROWN. In detail, in both *CIFAR_biasfield* and *CIFAR_tinyimagenet*, we achieved higher verified accuracy without incurring any *unknown* answer and with significantly less verification time. These strong final results demonstrate the effectiveness and impact of using `PT-LiRPA` for verification, showing the advantage of incorporating probabilistically optimal intermediate bounds in handling challenging instances that are difficult to solve with provable solvers.

## 5 CONCLUSION

We introduced `PT-LiRPA`, a novel probabilistic method that enhances the formal verification of deep neural networks by combining existing linear relaxation-based perturbation approaches with a sampling-based technique. Our approach provides tighter linear bounds while maintaining provable guarantees on the soundness of the result returned, significantly improving both the accuracy and computational efficiency of verification. Moreover, we presented a new analytical formula, $\Delta^*$, which offers a dual assurance of optimality for LiRPA bounds and qualitative insights into the error margin of sample-based estimations. Empirical results demonstrate that `PT-LiRPA` outperforms existing methods, particularly in terms of verification time, while also successfully addressing previously unsolved instances. Inspiring future directions involves studying the impact of this novel approach for verification guarantees for other realistic tasks, such as deep reinforcement learning or explainability of AI models.

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

APPENDIX

## A   PROOF OF LEMMA 2

**Lemma 2** (ReLU Layer Relaxation using `PT-LiRPA`). *Fix $\alpha, R \in (0, 1)$. Given two vectors $\boldsymbol{A}^*, \boldsymbol{v} \in \mathbb{R}^d$, where $\boldsymbol{v}$ is the pre-activated ReLU vector, with $\bar{\boldsymbol{l}} \leq \boldsymbol{v} \leq \underline{\boldsymbol{u}}$ (element-wise) obtained from a sampled of $n \geq \frac{\ln(1-\alpha)}{\ln(R)}$ samples and $\boldsymbol{A}^*$ is the vector of the linear bounds coefficients of the previous ReLU layer (computed with the probabilistically optimal intermediate bounds), with probability $\geq \alpha$ it holds:*

$$\boldsymbol{A}^{*T} ReLU(\boldsymbol{v}) \geq \boldsymbol{A}^{*T}(\underline{\boldsymbol{D}}^* \boldsymbol{v} + \underline{\boldsymbol{b}}^*) \tag{6}$$

*where*

$$\underline{\boldsymbol{D}}^* = \begin{cases} 1 & \bar{l}_j \geq 0, \\ 0 & \underline{u}_j \leq 0, \\ \alpha_j & \underline{u}_j > 0 > \bar{l}_j \text{ and } \boldsymbol{A}_j^* \geq 0, \\ \frac{\underline{u}_j}{\underline{u}_j - \bar{l}_j} & \underline{u}_j > 0 > \bar{l}_j \text{ and } \boldsymbol{A}_j^* < 0 \end{cases} \qquad \underline{\boldsymbol{b}}^* = \begin{cases} 0 & \bar{l}_j > 0 \text{ or } \underline{u}_j \leq 0, \\ 0 & \underline{u}_j > 0 > \bar{l}_j \text{ and } \boldsymbol{A}_j^* \geq 0, \\ -\frac{\underline{u}_j \bar{l}_j}{\underline{u}_j - \bar{l}_j} & \underline{u}_j > 0 > \bar{l}_j \text{ and } \boldsymbol{A}_j^* < 0. \end{cases}$$

*Proof.* We want to show that with a confidence $\geq \alpha$ we can still bound each ReLU layer even using $\underline{\boldsymbol{D}}^*$ and $\underline{\boldsymbol{b}}^*$, i.e., the new diagonal matrix, a bias vector computed exploiting the probabilistically optimal intermediate bounds.

To lower $\boldsymbol{A}^{*T} ReLU(\boldsymbol{v}) = \sum_j \boldsymbol{A}_j^*(ReLU(\boldsymbol{v}_j))$ we show that following the construction of $\underline{\boldsymbol{D}}^*$ and $\underline{\boldsymbol{b}}^*$, each term in the summation is probabilistically soundly bounded. Since the construction is the same as in any LiRPA approaches, we only replace $\boldsymbol{A}_j$ with $\boldsymbol{A}_j^*$ which is the $j$-th coefficient of the row vector $\boldsymbol{A}^*$ the encodes the linear relation with the previous layer, and $[\boldsymbol{l}_j, \boldsymbol{u}_j]$ the overestimated intermediate bounds computed, for instance via IBP, with $[\bar{\boldsymbol{l}}_j, \underline{\boldsymbol{u}}_j]$ which are the underestimated probabilistically optimal lower and upper bounds of the $j$-th node, computed with the sampling-based approach.

We start by noticing that for unstable ReLU nodes, the following inequality holds

$$\boldsymbol{l}_j \leq \boldsymbol{l}_j^* \leq \bar{\boldsymbol{l}}_j < 0 < \underline{\boldsymbol{u}}_j \leq \boldsymbol{u}_j^* \leq \boldsymbol{u}_j, \tag{7}$$

with $[\boldsymbol{l}_j^*, \boldsymbol{u}_j^*]$ the real lower and upper bound for that specific $j$-th node.

We prove the lemma by cases.

$\boxed{\underline{\boldsymbol{u}}_j > 0 > \bar{\boldsymbol{l}}_j}$

From inequality 7, we know that since we are underestimating true bounds $[\boldsymbol{l}_j^*, \boldsymbol{u}_j^*]$, the ReLU node is actually unstable, even for any LiRPA approach. Comparing the diagonal coefficients $\frac{\underline{\boldsymbol{u}}_j}{\underline{\boldsymbol{u}}_j - \bar{\boldsymbol{l}}_j}$ with $\frac{\boldsymbol{u}_j}{\boldsymbol{u}_j - \boldsymbol{l}_j}$ and the biases $-\frac{\underline{\boldsymbol{u}}_j \bar{\boldsymbol{l}}_j}{\underline{\boldsymbol{u}}_j - \bar{\boldsymbol{l}}_j}$ with $-\frac{\boldsymbol{u}_j \boldsymbol{l}_j}{\boldsymbol{u}_j - \boldsymbol{l}_j}$ of `PT-LiRPA` and any LiRPA cannot be helpful. The relation between the coefficients strongly depends on the quality of the bounds computed, and we cannot draw any direct conclusion since in some cases $\underline{\boldsymbol{D}} > \underline{\boldsymbol{D}}^*$ and in some cases not. Hence, we need to proceed by subcases.

$\underline{\boldsymbol{A}_j^* < 0}$. If $\boldsymbol{A}_j^* < 0$ the relation $\boldsymbol{A}_j^*(ReLU(\boldsymbol{v}_j)) \geq \boldsymbol{A}_j^*(\underline{\boldsymbol{D}}_{j,j}^* \boldsymbol{v}_j + \underline{\boldsymbol{b}}_j^*)$ to prove becomes $ReLU(\boldsymbol{v}_j) \leq \underline{\boldsymbol{D}}_{j,j}^* \boldsymbol{v}_j + \underline{\boldsymbol{b}}_j^*$ where if $\boldsymbol{v}_j < 0$ we have:

$$0 \leq \frac{\underline{\boldsymbol{u}}_j}{\underline{\boldsymbol{u}}_j - \bar{\boldsymbol{l}}_j} \boldsymbol{v}_j + \left( -\frac{\underline{\boldsymbol{u}}_j \bar{\boldsymbol{l}}_j}{\underline{\boldsymbol{u}}_j - \bar{\boldsymbol{l}}_j} \right) = \frac{\underline{\boldsymbol{u}}_j (\boldsymbol{v}_j - \bar{\boldsymbol{l}}_j)}{\underline{\boldsymbol{u}}_j - \bar{\boldsymbol{l}}_j}$$

since $\bar{\boldsymbol{l}}_j < 0$ and $\bar{\boldsymbol{l}}_j \leq \boldsymbol{v}_j$, thus $\boldsymbol{v}_j - \bar{\boldsymbol{l}}_j \geq 0$ which is enough to prove the inequality.

If $\boldsymbol{v}_j \geq 0$ we have:

$$\boldsymbol{v}_j \leq \frac{\underline{\boldsymbol{u}}_j}{\underline{\boldsymbol{u}}_j - \bar{\boldsymbol{l}}_j} \boldsymbol{v}_j + \left( - \frac{\underline{\boldsymbol{u}}_j \bar{\boldsymbol{l}}_j}{\underline{\boldsymbol{u}}_j - \bar{\boldsymbol{l}}_j} \right)$$

$$0 \leq \frac{\underline{\boldsymbol{u}}_j (\boldsymbol{v}_j - \bar{\boldsymbol{l}}_j)}{\underline{\boldsymbol{u}}_j - \bar{\boldsymbol{l}}_j} - \boldsymbol{v}_j$$

$$0 \leq \frac{\bar{\boldsymbol{l}}_j (\boldsymbol{v}_j - \underline{\boldsymbol{u}}_j)}{\underline{\boldsymbol{u}}_j - \bar{\boldsymbol{l}}_j}$$

where $\bar{\boldsymbol{l}}_j < 0$ and $(\boldsymbol{v}_j - \underline{\boldsymbol{u}}_j) < 0$. The inequality necessarily holds since the numerator and the denominator are positive. This concludes the first subcase.

$\underline{\boldsymbol{A}_j^* \geq 0}$. Now the condition to verify is $ReLU(\boldsymbol{v}_j) \geq \underline{\boldsymbol{D}}_{j,j}^* \boldsymbol{v}_j + \underline{\boldsymbol{b}}_j^*$. If $\boldsymbol{v}_j < 0$ we have

$$0 \geq \alpha_j \boldsymbol{v}_j + 0$$

since $0 < \alpha < 1$ and $\boldsymbol{v}_j$ negative, the inequality holds. Similarly if $\boldsymbol{v}_j \geq 0$ we have

$$\boldsymbol{v}_j \geq \alpha_j \boldsymbol{v}_j$$

which is clearly true. This concludes the first case.

For the next cases, $\bar{\boldsymbol{l}}_j > 0$ and $\underline{\boldsymbol{l}}_j > 0$, from proposition 1, we could define a ReLU node as (un)stable when, in reality, it is not. However, from lemma 1, we know that in even a potentially infinite sampling of points with a confidence $\alpha$ at most $(1 - R)$ points could fall outside of reachable set $[\bar{\boldsymbol{l}}_j, \underline{\boldsymbol{u}}_j]$. Thus, we can show the following cases with probability $\geq \alpha$.

$\boxed{\bar{\boldsymbol{l}}_j > 0}$

$\underline{\boldsymbol{A}_j^* < 0}$. We need to show that $ReLU(\boldsymbol{v}_j) \leq \underline{\boldsymbol{D}}_{j,j}^* \boldsymbol{v}_j + \underline{\boldsymbol{b}}_j^*$. Since $\bar{\boldsymbol{l}}_j > 0$ and $\bar{\boldsymbol{l}}_j \leq \boldsymbol{v}_j$ we have $\boldsymbol{v}_j \leq 1 \cdot \boldsymbol{v}_j + 0$ which is clearly true.

$\underline{\boldsymbol{A}_j^* \geq 0}$. We need to show that $ReLU(\boldsymbol{v}_j) \geq \underline{\boldsymbol{D}}_{j,j}^* \boldsymbol{v}_j + \underline{\boldsymbol{b}}_j^*$, which for similar previous consideration we have $\boldsymbol{v}_j \geq 1 \cdot \boldsymbol{v}_j$. This concludes the second case.

$\boxed{\underline{\boldsymbol{u}}_j < 0}$

$\underline{\boldsymbol{A}_j^* < 0}$. We need to show that $ReLU(\boldsymbol{v}_j) \leq \underline{\boldsymbol{D}}_{j,j}^* \boldsymbol{v}_j + \underline{\boldsymbol{b}}_j^*$. Since $\underline{\boldsymbol{u}}_j < 0$ and $\boldsymbol{v}_j \leq \underline{\boldsymbol{u}}_j$ we have $0 \leq 0$ which is clearly true.

$\underline{\boldsymbol{A}_j^* \geq 0}$. We need to show that $ReLU(\boldsymbol{v}_j) \geq \underline{\boldsymbol{D}}_{j,j}^* \boldsymbol{v}_j + \underline{\boldsymbol{b}}_j^*$, which for similar previous consideration we have $0 \geq 0$. This concludes the last case.

Hence we prove that with probability $\geq \alpha$ each term in the summation is soundly bounded by $\boldsymbol{A}^{*T} (\underline{\boldsymbol{D}}^* \boldsymbol{v} + \underline{\boldsymbol{b}}^*)$ thus concluding the argument.

$\square$

## B  EXAMPLE OF LINEAR COMPUTATION WITH LIRPA AND PT−LIRPA

In the following, we provide a simple example of linear bound computation for a toy DNN depicted in Figure 3. The neural network comprises two inputs, two hidden layers with ReLU activation, and one single output.

Following the notation introduced in §2 we define

$$\boldsymbol{W}^{(1)} = \begin{bmatrix} 2 & 1 \\ -3 & 4 \end{bmatrix}, \quad \boldsymbol{W}^{(2)} = \begin{bmatrix} 4 & -2 \\ 2 & 1 \end{bmatrix}, \quad \boldsymbol{w}^{(3)T} = [-2, 1];$$

and we set the bias terms in the layers to zero. We consider an original input $\boldsymbol{x}_0^T = [0, 1]$ and an $\ell_\infty \, \varepsilon = 2$ perturbation around it, thus obtaining a perturbation region $\mathcal{C} = [[-2, 2], [-1, 3]]$.

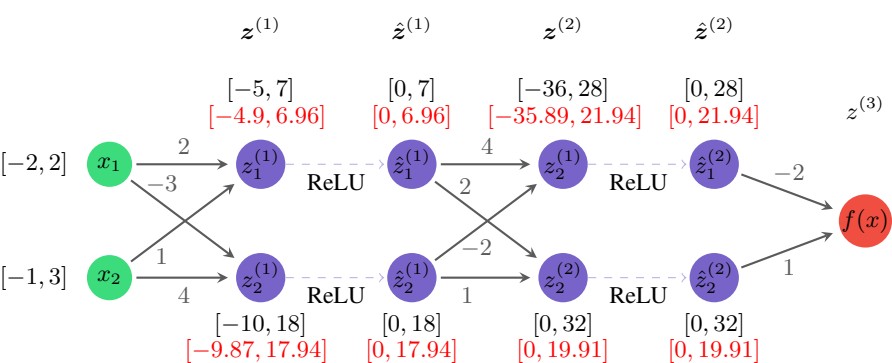

Figure 3: Toy DNN used in this example. Intervals reported in black are the result of the IBP for the input [[-2,2], [-1,3]]. In red, intermediate reachable sets are computed using a sampling-based approach in `PT-LiRPA`.

By propagating these intervals through the DNN, we obtain the interval $[-56, 32]$ as the output reachable set. Given the reasonable size of the neural network, before computing the linear lower and upper bounds using `LiRPA` and `PT-LiRPA`, we employed a MIP (Tjeng et al., 2017) solution to compute the true min and max of the function, respectively, which correspond to $[-32.53, 18.86]$.

To compute the lower and upper bound using `LiRPA`'s backward computation, we employ the CROWN (Zhang et al., 2018) strategy. To this end, it is useful to represent the neural network as reported in Figure 4.

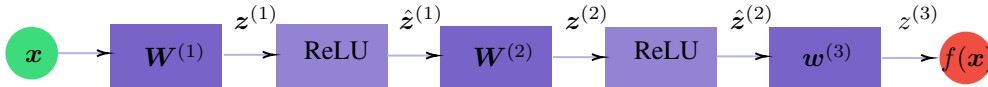

Figure 4: Alternative representation of toy DNN of Figure 3.

We note that $\hat{z}^{(2)}$ and $\hat{z}^{(1)}$ contain non-linear activation functions (ReLU), and we have to linearize them to keep the linear relationship between the output and these hidden layers. To this end, we can create a diagonal matrix $\underline{D}^{(2)}, \overline{D}^{(2)}, \underline{D}^{(1)}, \overline{D}^{(1)}$ and bias vectors $\underline{b}^{(2)}, \overline{b}^{(2)}, \underline{b}^{(1)}, \overline{b}^{(1)}$ reflecting the impact of ReLU nodes on the final output. We report for simplicity here the original definition provided in (Zhang et al., 2018) also reported in §2 (a similar definition is applied to compute the $i$-th layer $\overline{D}^{(i)}$ and $\overline{b}^{(i)}$ by switching the unstable case's checking conditions on $A_j$):

$$\underline{D}^{(i)} = \begin{cases} 1 & l_j \geq 0, \\ 0 & u_j \leq 0, \\ \alpha_j & u_j > 0 > l_j \text{ and } A_j^{(i+1)} \geq 0, \\ \frac{u_j}{u_j - l_j} & u_j > 0 > l_j \text{ and } A_j^{(i+1)} < 0 \end{cases}$$

$$\underline{b}^{(i)} = \begin{cases} 0 & l_j > 0 \text{ or } u_j \leq 0, \\ 0 & u_j > 0 > l_j \text{ and } A_j^{(i+1)} \geq 0, \\ -\frac{u_j l_j}{u_j - l_j} & u_j > 0 > l_j \text{ and } A_j^{(i+1)} < 0. \end{cases}$$

In the following, for simplicity, we always set $\alpha_j = 0$. Moreover, after defining the $i$-th diagonal matrix, we can also compute the $i$-th layer relaxation with respect to the output as $\underline{A}^{(i-1)} = \underline{A}^{(i)} \underline{D}^{(i-1)} W^{(i-1)}$ and similarly for the $\overline{A}^{(i-1)}$. In the beginning, we set $\underline{A}^{(4)} = \overline{A}^{(4)} = I$ and $\underline{A}^{(3)} = \overline{A}^{(3)} = w^{(3)T}$ and write starting from right to left (backward computation)[4]

---

[4]We report the lower bound version but for the upper we have similar consideration with the reversed inequality.

$$f(x) = z^{(3)}(\boldsymbol{x})$$

$$= \boldsymbol{w}^{(3)T} \hat{z}^{(2)}(\boldsymbol{x})$$

$$\geq \underline{\boldsymbol{A}}^{(3)} \underline{\boldsymbol{D}}^{(2)} z^{(2)}(\boldsymbol{x}) \qquad \text{computing a linearization for } \hat{z}^{(2)}$$

$$\geq \underbrace{\underline{\boldsymbol{A}}^{(3)} \underline{\boldsymbol{D}}^{(2)} \boldsymbol{W}^{(2)}}_{\underline{\boldsymbol{A}}^{(2)}} \hat{z}^{(1)}(\boldsymbol{x}) \qquad \text{rewriting } z^{(2)} = \boldsymbol{W}^{(2)} \hat{z}^{(1)}$$

$$\geq \underline{\boldsymbol{A}}^{(2)} \underline{\boldsymbol{D}}^{(1)} z^{(1)}(\boldsymbol{x}) \qquad \text{computing a linear bound for } \hat{z}^{(1)}$$

$$\geq \underbrace{\underline{\boldsymbol{A}}^{(2)} \underline{\boldsymbol{D}}^{(1)} \boldsymbol{W}^{(1)}}_{\underline{\boldsymbol{A}}^{(1)}} (\boldsymbol{x}) \qquad \text{rewriting } z^{(1)} = \boldsymbol{W}^{(1)} \hat{z}^{(0)} = \boldsymbol{W}^{(1)}(\boldsymbol{x})$$

$$\geq \underline{\boldsymbol{A}}^{(1)}(\boldsymbol{x}) + \underline{d}.$$

Hence, in order to linearize $\hat{z}^{(2)}(\boldsymbol{x})$ we compute $\underline{\boldsymbol{D}}^{(2)}, \overline{\boldsymbol{D}}^{(2)}$ and $\underline{\boldsymbol{b}}^{(2)}, \overline{\boldsymbol{b}}^{(2)}$ which presely correspond to

$$\underline{\boldsymbol{D}}^{(2)} = \begin{bmatrix} \frac{u}{u-l} & 0 \\ 0 & 1 \end{bmatrix} = \begin{bmatrix} 0.4375 & 0 \\ 0 & 1 \end{bmatrix} \qquad\qquad \underline{\boldsymbol{b}}^{(2)} = \begin{bmatrix} \frac{-ul}{u-l} \\ 0 \end{bmatrix} = \begin{bmatrix} 15.75 \\ 0 \end{bmatrix}$$

$$\overline{\boldsymbol{D}}^{(2)} = \begin{bmatrix} \alpha & 0 \\ 0 & 1 \end{bmatrix} = \begin{bmatrix} 0 & 0 \\ 0 & 1 \end{bmatrix} \qquad\qquad \overline{\boldsymbol{b}}^{(2)} = \begin{bmatrix} 0 \\ 0 \end{bmatrix}$$

where $\underline{\boldsymbol{D}}_{j,j}^{(2)}$ element is computed looking at each intermediate pre-activated bounds of $\boldsymbol{z}_j^{(2)}$ and the sign of $j$-th element of the vector $\underline{\boldsymbol{A}}^{(3)}$. Thus we have $\underline{\boldsymbol{A}}^{(2)} = \underline{\boldsymbol{A}}^{(3)} \underline{\boldsymbol{D}}^{(2)} \boldsymbol{W}^{(2)} = [-1.5, 2.75]$ and $\overline{\boldsymbol{A}}^{(2)} = \overline{\boldsymbol{A}}^{(3)} \overline{\boldsymbol{D}}^{(2)} \boldsymbol{W}^{(2)} = [2, 1]$. We proceed computing the diagonal matrix $\underline{\boldsymbol{D}}^{(1)}, \overline{\boldsymbol{D}}^{(1)}$ and bias vectors $\underline{\boldsymbol{b}}^{(1)}, \overline{\boldsymbol{b}}^{(1)}$ for $\hat{z}^{(1)}$. In detail, we obtain,

$$\underline{\boldsymbol{D}}^{(1)} = \begin{bmatrix} \frac{u}{u-l} & 0 \\ 0 & \alpha \end{bmatrix} = \begin{bmatrix} 0.583 & 0 \\ 0 & 0 \end{bmatrix} \qquad\qquad \underline{\boldsymbol{b}}^{(1)} = \begin{bmatrix} \frac{-ul}{u-l} \\ 0 \end{bmatrix} = \begin{bmatrix} 2.92 \\ 0 \end{bmatrix}$$

$$\overline{\boldsymbol{D}}^{(1)} = \begin{bmatrix} \frac{u}{u-l} & 0 \\ 0 & \frac{u}{u-l} \end{bmatrix} = \begin{bmatrix} 0.583 & 0 \\ 0 & 0.643 \end{bmatrix} \qquad\qquad \overline{\boldsymbol{b}}^{(1)} = \begin{bmatrix} \frac{-ul}{u-l} \\ \frac{-ul}{u-l} \end{bmatrix} = \begin{bmatrix} 2.92 \\ 6.43 \end{bmatrix}$$

with $\underline{\boldsymbol{A}}^{(1)} = \underline{\boldsymbol{A}}^{(2)} \underline{\boldsymbol{D}}^{(1)} \boldsymbol{W}^{(1)} = [-1.75, -0.875]$ and $\overline{\boldsymbol{A}}^{(1)} = \overline{\boldsymbol{A}}^{(2)} \overline{\boldsymbol{D}}^{(1)} \boldsymbol{W}^{(1)} = [0.40, 3.74]$.

Finally, we compute the sum if the bias vectors $\underline{d} = \underline{\boldsymbol{A}}^{(3)} \underline{\boldsymbol{b}}^{(2)} + \underline{\boldsymbol{A}}^{(2)} \underline{\boldsymbol{b}}^{(1)} = -35.88$ and $\overline{d} = \overline{\boldsymbol{A}}^{(3)} \overline{\boldsymbol{b}}^{(2)} + \overline{\boldsymbol{A}}^{(2)} \overline{\boldsymbol{b}}^{(1)} = 12.27$.

The final linear relation is thus $\underline{f}(\boldsymbol{x}) \geq \underline{\boldsymbol{A}}^{(1)}(\boldsymbol{x}) + \underline{d}$ and $\overline{f}(\boldsymbol{x}) \leq \overline{\boldsymbol{A}}^{(1)}(\boldsymbol{x}) + \overline{d}$. To compute the linear lower bound $f$ from this linear relation when $\mathcal{C}$ in an $\ell_\infty$ norm ball around $\boldsymbol{x}_0$, as in this example, can be easily obtained using Hölder's inequality (Zhang et al., 2018). In fact, we have

$$\underline{f}_{\text{CROWN}} = \min_{\boldsymbol{x} \in \mathcal{C}} \underline{\boldsymbol{A}}^{(1)}(\boldsymbol{x}) + \underline{d} = -||\underline{\boldsymbol{A}}^{(1)}||_1 \cdot \varepsilon + \underline{\boldsymbol{A}}^{(1)} \boldsymbol{x}_0 + \underline{d}$$

$$= -5.25 - 0.875 - 35.88 = -42.$$

$$\overline{f}_{\text{CROWN}} = \max_{\boldsymbol{x} \in \mathcal{C}} \overline{\boldsymbol{A}}^{(1)}(\boldsymbol{x}) + \overline{d} = ||\underline{\boldsymbol{A}}^{(1)}||_1 \cdot \varepsilon + \overline{\boldsymbol{A}}^{(1)} \boldsymbol{x}_0 + \overline{d}$$

$$= 8.28 + 3.74 + 12.27 = 24.29.$$

## B.1 PT−LiRPA COMPUTATION

The computation in PT−LiRPA is very similar to what we see above, with the exception of the construction of the diagonal matrices and bias vectors. In detail, we start by computing the prob-

abilistically optimal intermediate bounds from a sample-based approach in $\mathcal{C}$. We report in Figure 3 highlighted in red the results obtained from the propagation of $n$ random samples drawn from $[[-2, 2], [-1, 3]]$. As we can notice, the bounds are slightly tighter than the overestimated ones obtained from the interval bound propagation. Our intuition is thus that from the computation of $\underline{\boldsymbol{D}}^{(i)}, \overline{\boldsymbol{D}}^{(i)}, \underline{\boldsymbol{b}}^{(i)}, \overline{\boldsymbol{b}}^{(i)}$ using this tightened bounds we can obtain more accurate lower and upper final linear bounds. Thus we obtain:

$$\underline{\boldsymbol{D}}^{(2)} = \begin{bmatrix} \frac{u}{u-l} & 0 \\ 0 & 1 \end{bmatrix} = \begin{bmatrix} 0.3793 & 0 \\ 0 & 1 \end{bmatrix} \qquad \underline{\boldsymbol{b}}^{(2)} = \begin{bmatrix} \frac{-ul}{u-l} \\ 0 \end{bmatrix} = \begin{bmatrix} 13.6162 \\ 0 \end{bmatrix}$$

$$\overline{\boldsymbol{D}}^{(2)} = \begin{bmatrix} \alpha & 0 \\ 0 & 1 \end{bmatrix} = \begin{bmatrix} 0 & 0 \\ 0 & 1 \end{bmatrix} \qquad \overline{\boldsymbol{b}}^{(2)} = \begin{bmatrix} 0 \\ 0 \end{bmatrix},$$

and

$$\underline{\boldsymbol{D}}^{(1)} = \begin{bmatrix} \frac{u}{u-l} & 0 \\ 0 & \alpha \end{bmatrix} = \begin{bmatrix} 0.5868 & 0 \\ 0 & 0 \end{bmatrix} \qquad \underline{\boldsymbol{b}}^{(1)} = \begin{bmatrix} \frac{-ul}{u-l} \\ 0 \end{bmatrix} = \begin{bmatrix} 2.876 \\ 0 \end{bmatrix}$$

$$\overline{\boldsymbol{D}}^{(1)} = \begin{bmatrix} \frac{u}{u-l} & 0 \\ 0 & \frac{u}{u-l} \end{bmatrix} = \begin{bmatrix} 0.5868 & 0 \\ 0 & 0.6451 \end{bmatrix} \qquad \overline{\boldsymbol{b}}^{(1)} = \begin{bmatrix} \frac{-ul}{u-l} \\ \frac{-ul}{u-l} \end{bmatrix} = \begin{bmatrix} 2.876 \\ 6.367 \end{bmatrix}.$$

We can now compute all the $\boldsymbol{A}$s and $d$s vectors.

$$\underline{\boldsymbol{A}}^{(2)} = \underline{\boldsymbol{A}}^{(3)} \underline{\boldsymbol{D}}^{(2)} \boldsymbol{W}^{(2)} = [-1.0351, 2.517]$$

$$\overline{\boldsymbol{A}}^{(2)} = \overline{\boldsymbol{A}}^{(3)} \overline{\boldsymbol{D}}^{(2)} \boldsymbol{W}^{(2)} = [2, 1]$$

$$\underline{\boldsymbol{A}}^{(1)} = \underline{\boldsymbol{A}}^{(2)} \underline{\boldsymbol{D}}^{(1)} \boldsymbol{W}^{(1)} = [-1.2149, -0.6074]$$

$$\overline{\boldsymbol{A}}^{(1)} = \overline{\boldsymbol{A}}^{(2)} \overline{\boldsymbol{D}}^{(1)} \boldsymbol{W}^{(1)} = [0.4121, 3.7541]$$

$$\underline{d} = \underline{\boldsymbol{A}}^{(3)} \underline{\boldsymbol{b}}^{(2)} + \underline{\boldsymbol{A}}^{(2)} \underline{\boldsymbol{b}}^{(1)} = -30.209$$

$$\overline{d} = \overline{\boldsymbol{A}}^{(3)} \overline{\boldsymbol{b}}^{(2)} + \overline{\boldsymbol{A}}^{(2)} \overline{\boldsymbol{b}}^{(1)} = 12.119$$

Finally we have

$$\underline{f}_{\text{PT-LiRPA}} = \min_{\boldsymbol{x} \in \mathcal{C}} \underline{\boldsymbol{A}}^{(1)}(\boldsymbol{x}) + \underline{d} = -||\underline{\boldsymbol{A}}^{(1)}||_1 \cdot \varepsilon + \underline{\boldsymbol{A}}^{(1)} \boldsymbol{x}_0 + \underline{d}$$
$$= -3.6447 - 0.6074 - 30.209 = -34.46.$$

$$\overline{f}_{\text{PT-LiRPA}} = \max_{\boldsymbol{x} \in \mathcal{C}} \overline{\boldsymbol{A}}^{(1)}(\boldsymbol{x}) + \overline{d} = ||\underline{\boldsymbol{A}}^{(1)}||_1 \cdot \varepsilon + \overline{\boldsymbol{A}}^{(1)} \boldsymbol{x}_0 + \overline{d}$$
$$= 8.33 + 3.7541 + 12.119 = 24.20.$$

As we can notice, even in this toy example, our procedure produces tighter bounds compared to the original CROWN approach, confirming the correctness of our hypothesis.

## C EMPIRICAL EVALUATION: FURTHER DETAILS

All the data are collected on a cluster running Rocky Linux 9.34 equipped with Nvidia RTX A6000 (48 GiB) and a CPU AMD Epyc 7313 (16 cores). To test the scalability and effectiveness of PT-LiRPA, in the first set of experiments, we consider different non-linear activation functions, such as Tanh and Sigmoid, beyond ReLU and models of different sizes. We report in Table 4 details on the input and hidden sizes of the models tested. For larger models, since MIP cannot be exploited, we employ the Powell algorithm (Powell, 1989) implemented in SciPy (Virtanen et al., 2020) to still have an intuition of the tightness of the computed bounds in terms of $\ell_2$ norm. However, since

| Hyperparameters first part empirical evaluation | | | | | | |
|---|---|---|---|---|---|---|
| Model tested | Input size; domain | $\varepsilon$ perturbation | hidden sizes; depth | Activation functions | LiRPA bound prop. | `PT-LiRPA` hyperparams |
| *500* | [2]; [0, 1] | [1, 2] | [1, 2]; [32] | ReLU | CROWN, $\alpha$-CROWN (*optimized* iterations:20,lr_alpha:0.1) | $\alpha \geq 99.99\%$ $R = 0.001$ $n = 10k$ |
| *1500* | [2,4]; [0, 1] | [1, 2] | [2,4]; [32, 64, 128] | ReLU | CROWN, $\alpha$-CROWN (*optimized* iterations:20,lr_alpha:0.1) | $\alpha \geq 99.99\%$ $R = 0.001$ $n = 10k$ |

Table 4: Hyperparameters used for the first set of experiments.

this algorithm solves the minimization problem in terms of local minima, before computing the $\ell_2$ norm and the $\Delta^*$, we first check if both LiRPA and `PT-LiRPA` produce valid lower bounds ensuring that LiRPA and `PT-LiRPA` produce linear bounds that are smaller (and greater for `PT-OVER`) than the Powell minima, respectively. Results reported in Tab. 1 and 2, confirm the correctness of Lemma 3 and our intuition on the tightness of linear bounds when using probabilistically optimal intermediate reachable sets. Specifically, our empirical results on 2000 models `PT-LiRPA` always produced a valid lower bound of $f(x)$, comparing the minimum discovered by our approach and the one returned by MIP and Powell.

**Further details on the benchmark employed in the verification test.** All the benchmarks used in our empirical evaluation are comprehensively discussed in (Müller et al., 2022; Brix et al., 2023). To keep the paper self-contained, we report below a brief overview of the selected benchmarks.

- *ACAS xu* (Julian et al., 2016; Katz et al., 2017) benchmark 2023: includes ten properties evaluated across 45 neural networks designed to provide turn advisories for aircraft to prevent collisions. Each neural network consists of 300 neurons distributed over six layers, using ReLU activation functions. The networks take five inputs representing the aircraft's state and produce five outputs, with the advisory determined by the minimum output value. Here, we verified only property 3, which returns unsafe if COC is minimal, with a max computation time of 116s.

- *TllVerifyBench* benchmark 2023: this benchmark features Two-Level Lattice (TLL) neural networks with two input and one single output. These models are then transformed into MLP ReLU networks where the output properties consist of a randomly generated real number and a randomly generated inequality direction to be verified. Here we verify all 32 instances of the VNN-COMP 2023 with a timeout of 600s for each property.

- *CIFAR_biasfield* benchmark 2022: this benchmark focuses on verifying a Cifar-10 network under bias field perturbations. These perturbations are modeled by creating augmented networks that reduce the input space to just 16 parameters. For each image to be verified, a distinct bias field transform network is generated, consisting of a fully connected (FC) transform layer followed by the Cifar CNN with 8 convolutional layers with ReLU activations. Each bias field transform network has 363k parameters and 45k nodes. Here, we test all 72 properties with a timeout set to 300s for each one.

- *TinyImageNet* benchmark 2022: consists of CIFAR100 image classification ($56 \times 56 \times 3$) with Residual Neural Networks (ResNet). Here, we consider the medium network size composed of 8 residual blocks, 17 convolutional layers, and 2 linear layers. For *TinyImageNet-ResNet-medium*, we verify all 24 properties with a timeout of 200 seconds for each property.

In general, we selected benchmarks where the state-of-the-art $\alpha, \beta$-CROWN method is unable to solve some of the instances.

# D ABLATION STUDY

In this section, we study the impact of different sample sizes on the computation of the intermediate reachable sets.

Although Lemma 1 provides a lower bound on the number of samples needed to achieve a confidence level of $\alpha$ with an accuracy of at least $R$, we explore the effect of varying incremental sample sizes on

the computation of intermediate bounds. Specifically, we focus on the *CIFAR_biasfield* benchmark, which involves networks of substantial size. We begin with a confidence level of $\alpha \geq 99.9\%$ and set $R = 0.9995$, requiring 1,378 samples. As a stopping criterion for the experiment, we establish a distance threshold of $\Delta = 0.001$ between the intermediate reachable sets computed with the tested sample sizes and the reference bounds, which are determined using the maximum achievable sample size before encountering an *out of memory* error– in our settings $350k$ samples. Thus, we progressively increase the sample size until the discrepancy exceeds the threshold.

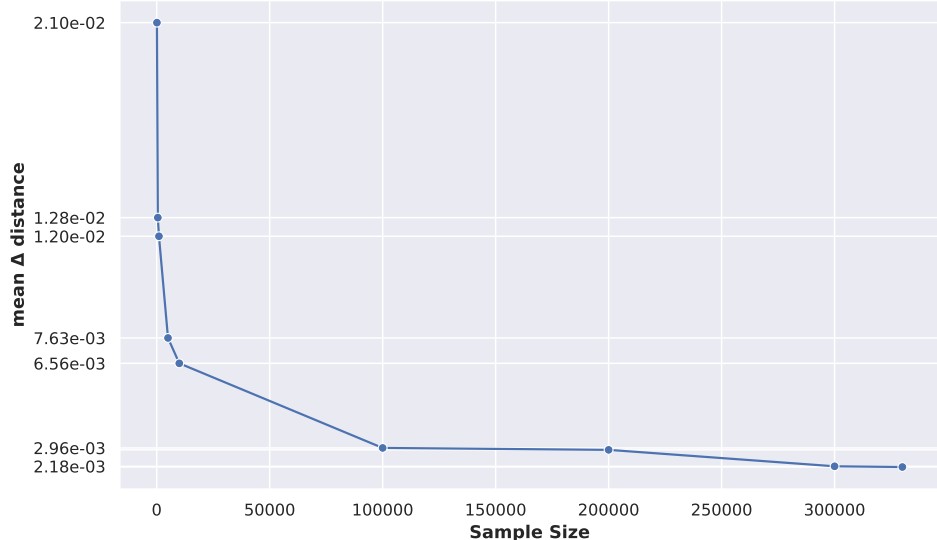

Figure 5: Intermediate bounds convergence for the increasing sample size in *CIFAR_biasfield* benchmark. $y$-axis reports the mean distance between intermediate bounds using $350k$ samples (as reference) and the one using $[100, 500, 1k, 5k, 10k, 100k, 200k, 300k, 330k]$, respectively.

Our results detailed in Figure 5 indicate that stable intermediate reachable sets, in this scenario, can be obtained with sample sizes ranging from $250k$ to $330k$ as the mean distance between intermediate bounds is strictly less than $\Delta = 0.001$. It is important to highlight that propagating a large number of samples, such as $350k$, requires a computational effort and time comparable to propagating significantly fewer samples due to batch processing and GPU acceleration. The primary limitation is the GPU's memory capacity, as higher sample sizes typically increase the likelihood of memory errors compared to the use of CPU propagation.

