# OpenReview forum: "Enhancing Linear Bound Tightness in Neural Network Verification via Sampling-Based Underestimation"
_ICLR.cc/2025/Conference — ICLR 2025 Conference Withdrawn Submission_

### Official Review · Reviewer_Kwx4 · 2024-10-17

**Soundness:** 3
**Presentation:** 2
**Contribution:** 2
**Rating:** 3
**Confidence:** 4

**Summary:**

This work introduces a probabilistic approach for tightening bounds in neural network verification which are computed using the popular CROWN algorithm. To obtain initial bounds for intermediate layers in a network, most works use algorithms which propagate concrete intervals through a network. The authors suggest to instead obtain probabilistic bounds for intermediate layers by sampling perturbations from the input set and running simple forward passes through the network, keeping track of the minimum and maximum values observed for each neuron in the network. These bounds may be unsound, but the authors derive theoretical guarantees on the confidence level with which the bounds hold based on the number of inputs sampled to obtain the bounds. They show that CROWN bounds computed using these initial bounds hold with the same confidence level as the initial bounds, and demonstrate that the tightened intermediate bounds lead to tighter CROWN bounds. The improved bounds are shown to accelerate branch-and-bound-based neural network verifiers and to increase the number of instances that they are able to verify.

**Strengths:**

- While sampling-based bounding methods have been proposed before, this work presents a probabilistic bounding method which is fast since it does not require an LP solver
- There are thorough proofs and examples in the appendix which are helpful for understanding the method
- The experimental results of the method seem promising when comparing its verification performance to abcrown

**Weaknesses:**

- My main concern regarding the paper is the lack of contextualisation and comparison with other literature in probabilistic verification. There are some papers which also provide safety certificates that hold with a given confidence level, for example [1, 2]. Since this work is effectively attempting to do the same, it should be compared to this other work, possibly also including approaches such as randomised smoothing.
- Based on the current experimental evaluation I find it hard to assess whether this tightening approach is useful or not. It is only compared to one baseline (abcrown with standard LiRPA). abcrown is a sound verifier while abcrown with PT-LiRPA is an unsound verifier (since with probability $1-\alpha$ it produces incorrect verification results). I would argue that it is to be expected that soundness comes at a price and an unsound verifier would therefore be faster. Experimental results comparing this method with other probabilistic verification methods would therefore be important. Besides this, a comparison with GCP-CROWN which uses cutting planes to tighten bounds and is also part of the abcrown package would be interesting. In my opinion, the paper is not ready for acceptance in its current state.
- The authors should consider adding experiments that vary the confidence level to show how performance in terms of runtime/bound tightness changes with varying confidence levels being used.
- The authors show that probabilistic tightening of intermediate bounds at a given confidence level results in the final output bounds computed via CROWN holding with the same confidence level. This means that a verification result obtained from e.g. abcrown using PT-LiRPA is not necessarily correct (since the intermediate bounds used to obtain it may have been unsound). This should be made clearer especially in the experimental section, e.g. in statements such as "We achieve higher verified accuracy without incurring any unknown answer and with significantly less verification time" since the verified accuracy by abcrown+PT-LiRPA is not necessarily "true" verified accuracy.
- The baseline used for obtaining intermediate layer bounds should be clarified - I think in the current state of the paper this can be a bit confusing. In line 199 the authors state that intermediate results are obtained using "bound propagation" but CROWN, forward symbolic interval propagation, (non-symbolic) interval propagation are all bound propagation methods and the authors themselves use "bound propagation" to refer to CROWN later in the paper. In the appendix it becomes clear that the default method used to compute initial bounds in auto_LiRPA is Interval Bound Propagation (IBP), I think it would make the paper easier to understand if the main part of the paper would state this more clearly.

## Literature
[1] M. Fazlyab, M. Morari and G. J. Pappas, "Probabilistic Verification and Reachability Analysis of Neural Networks via Semidefinite Programming," 2019 IEEE 58th Conference on Decision and Control (CDC), Nice, France, 2019, pp. 2726-2731, doi: 10.1109/CDC40024.2019.9029310.

[2] Weng, L., Chen, P., Nguyen, L., Squillante, M., Boopathy, A., Oseledets, I. & Daniel, L.. (2019). PROVEN: Verifying Robustness of Neural Networks with a Probabilistic Approach. Proceedings of the 36th International Conference on Machine Learning, in Proceedings of Machine Learning Research 97:6727-6736

## Editorial
There are a number of grammatical/spelling mistakes in the paper, I'll list some of the ones that I noticed below:
- Line 71: enables to verify in a sound (and sometimes complete) way DNNs --> **enables the verification of DNNs in a sound (and sometimes complete) way**
- Line 86: we show that --> we show that **when**
- Line 95: $\underline{f}_ \text{OVER}$ the brown dot represents --> $\underline{f}_ \text{OVER}$ **,** the brown dot **,** represents
- Line 97: By employing a similar procedure also to compute --> By **also** employing a similar procedure to compute
- Line 129: based on LiRPA counterpart --> based on **the** LiRPA counterpart
- Line 149f: We define the symbol (...) the pre and (...) --> We define the symbol (...) **as** the pre and (...)
- Line 150: We consider Rectified Linear Unit (ReLU) as an activation function --> We consider **the** Rectified Linear Unit (ReLU) as an activation function
- Line 151: in the literature verification works --> in the **verification literature**
- Line 153: functions studied in literature --> functions studied in **THE** literature
- Line 155: a single output node on whose we can verify --> a single output node **which we can use** to verify
- Line 182: searching tree --> **search** tree
- Line 195: by computing linear relaxation of each --> by computing **a** linear relaxation of each
- Line 203: In other cases, is either considered --> In other cases, **it** is either considered
- Line 249: last years VNN-COMP --> either last year's or last years'
- Line 280: perturbation set on interest --> perturbation set **of** interest
- Line 294: it necessary holds --> it **necessarily** holds
- Line 321: obtained from a sampled of (...) samples --> obtained from (...) samples
- Line 364: to search for potential adversarial input --> to search for potential adversarial input**s**
- Line 365: If no adversarial is found --> If no adversarial **example** is found
- Line 367: using get_interm_bounds method --> using **the** get_interm_bounds method
- Line 460: to have still an intuition of --> to **still** have an intuition of
- Line 461: compute the mean over all the models tested of our novel analytic formula --> compute the mean of our novel analytics formula **across all the models tested**
- Line 487 (in the table header): unkwown --> **unknown**
- Line 825: presely --> **precisely**

**Questions:**

- Line 534: What do the authors mean by "dual assurance of optimality for LiRPA bounds" which is provided by their new $\Delta^*$ formula?
- Line 420: We know for sure that $\underline{f}_ \text{OVER} \geq f^*$ but we do not know for sure whether $f_\text{PT-LiRPA} \leq f^*$ which means that we can have scenarios where $\Delta^* < 0$. In the experimental evaluation the confidence level and number of samples on which the algorithm is evaluated are chosen in a way which makes it unlikely for unsound bounds to occur. How should negative $\Delta^*$ generally be interpreted, does the proposed measure make sense in this case?
- Is there a reason why the method was compared to abcrown but not GCP-CROWN (which is the version of abcrown that uses cutting planes for bound tightening and which participated in VNNCOMP2022/2023/2024)?

---

### Official Review · Reviewer_n6m8 · 2024-10-29

**Soundness:** 1
**Presentation:** 2
**Contribution:** 1
**Rating:** 3
**Confidence:** 5

**Summary:**

The paper presents a novel algorithm to perform neural network verification: PT-LiRPA.
The core idea is to replace the intermediate bounds (bounds on the ranges of possible activation values at hidden layers, given a set of allowed inputs) obtained through network over-approximations by under-approximations obtained through uniform sampling.
These under-approximations are presented as probabilistic intermediate bounds.
The experiments present results obtained by integrating PT-LiRPA within popular neural network verification frameworks.

**Strengths:**

The idea of using probabilistic under-approximations is potentially interesting for setups where probabilistic certificates are considered to provide sufficient security guarantees.

**Weaknesses:**

In spite of a series of misleading claims from the authors (for instance, the abstract states that "*this approach preserves the soundness of verification results*"), there is no guarantee that the provided bounds on the verification problem (definition 1) are sound.
The authors (and this is particularly evident in the experiments) would appear to be comparing bounds that are deterministically sound with bounds that are sound only with a certain probability (those from PT-LiRPA). This is not a fair comparison by any means.
I think there is space for probabilistic guarantees in the area. However, these should be clearly marked as such throughout the motivational text, and compared to appropriate baselines. No probabilistic certification scheme (e.g., randomized smoothing) is even cited here. I think these points alone would justify rejection.

In addition, I think the way probabilistic guarantees of soundness are obtained is itself misleading.
The obtained intermediate bounds are an under-approximation by definition: in order to be sound, the exact global optimum (minimum for lower bounds, maximum for upper bounds) for the activation to bound needs to be within the sampled points.
In other words, in order for the bounds to be sound, a solution to a problem as hard as the original verification problem (over a subset of the network) must have been found through random sampling.
This way, the intermediate bounds are either exact, or unsound.

I do not think that the employed probabilistic machinery is appropriate for the purpose. As far as I understand, one could simply sample the network output directly, and --according to Lemma 1-- get a probabilistically sound lower bound on the network output. Given the high dimensionality of the input space, I believe it would be relatively easy to show that adversarial attacks (such as PGD or AutoAttack) can find tighter upper bounds to the verification problem.
The authors claim that no unsoundness was found in their experiments: I would speculate that this is due to the fact that they compose an over-approximation with an under-approximation (hence heuristically yielding bounds that are empirically valid).
In Lemma 1, any choice of $\alpha < 1$ implies that there is a non-null probability that the bounds are not sound for infinitely many points. I think that an appropriate probabilistic guarantee should be sound for *all* points in the input set with a given probability.

Finally, the novelty associated to the formula in section 3.2 is limited at best. Branch and bound frameworks will store an upper bound to the verification problem (for instance, obtained through adversarial attacks). If lower and upper bounds are less than an epsilon apart, the procedure terminates. This directly corresponds to evaluating a non-normalized version of equation (5). Similar measures are also commonly reported in experiments in the area: see for instance Figure 2 from the cited beta-CROWN paper (Wang et al., 2021).

**Questions:**

- The authors refer to their bounds as "probabilistically optimal". However, it is unclear in which sense the notion of optimality is employed.
- The authors should compare the obtained intermediate bounds with intermediate bounds under-approximations obtained through PGD. Repeating this for each intermediate bound and each experiment, it is extremely likely that (if run for enough steps), PGD will find a wider range for the activation, implying that the sampling-based range was not sound.
- The employed sample sizes are significant. Unless the network is extremely small, this must have a non-negligible impact on runtime. Could the authors provide measurements on networks with hundreds of thousands of neurons?
- I would think that Lemma 1 should be applied individually to each activation bound. If each bound is sound with a given probability $\alpha$, shouldn't the final bounds be sound only if all the intermediate bounds are sound, yielding a probability of $\alpha^{k}$, k being the number of bounds carried out?

---

### Official Review · Reviewer_JqnU · 2024-11-03

**Soundness:** 3
**Presentation:** 3
**Contribution:** 3
**Rating:** 5
**Confidence:** 4

**Summary:**

This paper proposes a novel approach called PT-LiRPA (Probabilistically Tightened Linear Relaxation-based Perturbation Analysis) to improve existing linear relaxation-bound calculation methods in neural network verification through sampling techniques. PT-LiRPA combines LiRPA methods with a sampling-based underestimation technique to calculate probabilistically optimal intermediate bounds, resulting in tighter linear upper and lower bounds. The study demonstrates that this approach preserves the reliability of verification results while significantly tightening bounds for nonlinear functions. Additionally, the authors introduce a new metric, ∆∗, to quantify the tightness of LiRPA bounds and assess the possible error in sample-based overestimations. Experimental results on various benchmark datasets show that PT-LiRPA outperforms or is comparable in verification accuracy and computational efficiency, allowing it to handle instances where previous methods failed.

**Strengths:**

-  Proposes an innovative approach to improve existing LiRPA methods by using sampling to create tighter bounds.
 - Provides theoretical guarantees for the reliability of the method at a given confidence level.

**Weaknesses:**

- The paper could benefit from more detailed explanations of the methodology. The evaluation is primarily focused on toy dataset as MNIST, CIFAR-10, still need more results for Tiny-ImagetNet and even for ImageNet Comparisons with a broader range of existing methods could provide a more comprehensive assessment of Ti-Lin's performance. The paper seems focus on CNN, wonder if this kind of method can also applied to Transformer-based model. The format of the numerical result should keep as the same.
- missing baseline [a1]
[a1]Xue, Zhiyi, et al. "A tale of two approximations: Tightening over-approximation for DNN robustness verification via under-approximation." Proceedings of the 32nd ACM SIGSOFT International Symposium on Software Testing and Analysis. 2023.
- missing time results in Table 1 and 2.

**Questions:**

- Could the authors explain why Dual-App[a1], which also uses under-estimation to enhance the precision of robustness verifaction , were not included in the comparisons?

[a1] Xue, Zhiyi, et al. "A tale of two approximations: Tightening over-approximation for DNN robustness verification via under-approximation." Proceedings of the 32nd ACM SIGSOFT International Symposium on Software Testing and Analysis. 2023.

- Could the authors include time efficiency results in Table 1 and 2? It would be helpful for readers to understand how much slower the approach is compared to CROWN.

- The paper seems focus on CNN. Could this method also be applied to Transformer-based models?

**Details Of Ethics Concerns:**

/

---

### Official Review · Reviewer_nUFZ · 2024-11-04

**Soundness:** 1
**Presentation:** 2
**Contribution:** 1
**Rating:** 3
**Confidence:** 4

**Summary:**

This paper replaces sound intermediate bounds in LiRPA for neural network verification by sampling-based estimations, for a probabilistic verification method.

**Strengths:**

* The paper proposed to develop a probabilistic verification method for neural networks by replacing the intermediate bounds in LiRPA with sampling-based estimations with provable probabilistic guarantees.
*  Results show that more instances in benchmarks can be “verified” under the probabilistic verification setting, and the results are compared to previous works under a deterministic verification setting.

**Weaknesses:**

* Probabilistic verification for neural networks appeared many years ago (no later than 2018). However, the authors did not make any comparison with existing neural network verification works which produce probabilistic guarantees (such as https://arxiv.org/abs/1812.08329, https://arxiv.org/abs/2405.17556) rather than deterministic guarantees.
* The current comparison in the paper is highly misleading: This paper essentially considers a different problem of neural network verification with probabilistic verification (with a confidence level) but the paper keeps comparing the proposed method with previous works on deterministic verification (100% guaranteed) and claiming that the proposed method on probabilistic verification has “outperformed” previous methods for deterministic verification. These two categories of methods are not comparable and such comparison is meaningless (when you make the guarantees less strict, of course you will get tighter bounds in the new setting).

**Questions:**

The authors need add many comparisons with existing works on probabilistic verification.

The authors should significantly revise the paper to make it clear (in both text and tables) that: 1) Their use of VNN-COMP benchmark has changed, because the original VNN-COMP benchmark is only for deterministic verification. 2) Guarantees generated by PT-LiRPA and deterministic baselines (such as α,β-CROWN) are different. These should be clearly marked in the tables.

---

### Note · Authors · 2024-11-28

**Comment:**

Dear Reviewers,

We appreciate the valuable comments and feedback provided on our submission. While we intended to convey that our solution incorporates a confidence-based perspective, we acknowledge the importance of comparing our method to other probabilistic verification approaches to strengthen its contribution.

We attempted to conduct such an experiment, however, the referenced works either lack complete implementations or do not provide the code needed for a straightforward comparison. To address this critical aspect, we have decided to withdraw the paper at this time. We aim to thoroughly cover this comparison and resubmit a more comprehensive version in the future.

Thank you for your time.
Sincerely,
The Authors

**Withdrawal Confirmation:**

I have read and agree with the venue's withdrawal policy on behalf of myself and my co-authors.